# Current Strategies for Tracheal Replacement: A Review

**DOI:** 10.3390/life11070618

**Published:** 2021-06-25

**Authors:** Giuseppe Damiano, Vincenzo Davide Palumbo, Salvatore Fazzotta, Francesco Curione, Giulia Lo Monte, Valerio Maria Bartolo Brucato, Attilio Ignazio Lo Monte

**Affiliations:** 1Department of Surgical, Oncological and Oral Sciences, University of Palermo, 90127 Palermo, Italy; giuseppe.damiano@unipa.it (G.D.); salvatore.fazzotta@unipa.it (S.F.); curionef@gmail.com (F.C.); attilioignazio.lomonte@unipa.it (A.I.L.M.); 2Euro-Mediterranean Institute of Science and Technology, 90139 Palermo, Italy; 3School of Biotechnology, University of Palermo, 90127 Palermo, Italy; panorgan@unipa.it; 4Department of Engineering, University of Palermo, 90127 Palermo, Italy; valerio.brucato@unipa.it

**Keywords:** bioengineering, biocompatible materials, scaffolds, surgical biotechnologies

## Abstract

Airway cancers have been increasing in recent years. Tracheal resection is commonly performed during surgery and is burdened from post-operative complications severely affecting quality of life. Tracheal resection is usually carried out in primary tracheal tumors or other neoplasms of the neck region. Regenerative medicine for tracheal replacement using bio-prosthesis is under current research. In recent years, attempts were made to replace and transplant human cadaver trachea. An effective vascular supply is fundamental for a successful tracheal transplantation. The use of biological scaffolds derived from decellularized tissues has the advantage of a three-dimensional structure based on the native extracellular matrix promoting the perfusion, vascularization, and differentiation of the seeded cell typologies. By appropriately modulating some experimental parameters, it is possible to change the characteristics of the surface. The obtained membranes could theoretically be affixed to a decellularized tissue, but, in practice, it needs to ensure adhesion to the biological substrate and/or glue adhesion with biocompatible glues. It is also known that many of the biocompatible glues can be toxic or poorly tolerated and induce inflammatory phenomena or rejection. In tissue and organ transplants, decellularized tissues must not produce adverse immunological reactions and lead to rejection phenomena; at the same time, the transplant tissue must retain the mechanical properties of the original tissue. This review describes the attempts so far developed and the current lines of research in the field of tracheal replacement.

## 1. Introduction

Airway diseases requiring surgical treatment have been increasing in recent years, due to the higher incidence of respiratory tract cancer (19% of tumors in Italy) [1]. Tracheal resections frequently cause disability, due to the loss of phonation and the presence of a permanent tracheostomy (that represents an easy access to pathogens responsible of recurrent lower respiratory tract infections, often lethal). Laryngotracheal stenosis is the main cause of re-intervention in the upper respiratory tract.

Tracheal resections inferior to 50% of the tracheal length are currently repaired by end-to-end anastomosis [2]. Any excessive tensions are approached by different techniques, including neck flexion, transcervical mobilization of the trachea and mainstem bronchi and the suprahyoid laryngeal release, transthoracic elevation of the right hilus with division of the pulmonary ligament, intrapericardial dissection of the pulmonary trunk, and incision of the tracheal intercartilaginous ligaments. The extent of tracheal resection length of depends on age, posture, and severity of disease [3,4,5].

Extensive lesions of the trachea, including tumors or stenoses overcoming half of the organ in adults (6 cm) or one third in children, may require prostheses [6].

The apparent simplicity of the “windpipe” encouraged studies with tubular substitutes including biosynthetic materials, auto- and allografts. The characteristics of an ideal tracheal substitute include lateral rigidity and longitudinal flexibility, and the presence of a ciliated respiratory epithelium covering the inner surface. The implant must be airtight and not produce chronic inflammation, excessive granulation tissue, infections, or erosions [7].

Materials should be biocompatible, nontoxic, non-immunogenic, and non-carcinogenic permitting epithelial covering. At the same time, materials should not migrate or producing erosion. Stenoses, increased secretions, and bacterial colonization are frequently observed phenomena [8]. This review describes the current approaches and progresses in the field of biosynthetic substitutes for tracheal replacement.

A systematic search of the literature electronic databases Web of Science, Scopus, and PubMed was performed. The search ranged from January 1947 to December 2020 by using the following keywords: tracheal reconstruction, artificial trachea, tracheal scaffolds, tracheal reconstruction, tracheal tissue regeneration. More than 3800 studies were identified. The selection was focused on whole tracheal reconstruction and included preclinical and clinical investigations. Twelve hundred papers were selected according to journal citations and indexing.

An electronic search on Clinicaltrial.gov was also performed with the same keywords, but only five results were available.

In Table 1, a summary of the main tracheal reconstruction techniques is reported.

## 2. Main Body

### 2.1. Anatomical Properties of the Trachea

The trachea is a flexible fibro-elastic tube that connect the larynx and the lungs [20]. The anterior and lateral sides consists of a pile of C-shaped rings of hyaline cartilage. Posterior wall is only covered with smooth muscle. The inner surface is internal with a pseudostratified columnar epithelium containing goblet cells. Tissue-engineering techniques for building a trachea require careful considerations of the anatomical layers: over epithelium and cartilage, there is a connective tissue that acts as a matrix for blood vessels and nerves, but it also has a supportive function for exocrine glands, an important source of mucus and fluids.

### 2.2. Standard Tracheal Reconstruction

Current trends for tracheal reconstruction include tracheal dilatation with rigid bronchoscope, laser surgery with placement of an endoluminal stent, and surgical resection. Unfortunately, the first two methods are burdened by very high recurrence of stenosis (90% for endoscopic dilation and 30–40% for laser treatment) [9]. Tracheal resection, therefore, represents the gold standard for the treatment of tracheal stenoses, especially in those cases in which the stenotic tract is lower than 4 cm in length [21]. Tracheal resection is usually carried out in primary tracheal tumors or other neoplasms such as thyroid carcinoma infiltrating the trachea [10], thymoma [22], or even in endotracheal post-intubation stenosis [23], in vocal cords palsy [24], tracheal atresia [25], or tracheo-esophageal fistula [26]. The post-operative period could be burdened by several complications (up to 20% of cases) that include recurrent stenosis, permanent tracheostomy, and even death [27].

### 2.3. Approaches to Tracheal Reconstruction

#### 2.3.1. Allografts

##### Tracheal Allografts

Fresh tracheal allografts have demonstrated early stenosis, necrosis, undergo liquefaction [2,11,28,29,30], and more important, develop graft rejection in the absence of immunosuppressive therapy. Tracheal allografts carry antigens [31] represented by HLA-DR expressed on epithelial cell surface. These antigens activate T lymphocytes and thereby trigger graft rejection. Chemical or physical treatments have also been proposed in clinical applications [4]. A high irradiation of allografts permits to avoid immunosuppression in the host, but the implant needs indirect vascular supply [32]. Chemical treatments do not induce rejection, but nope cartilage and epithelium formation [33]. Formalin-fixed allografts from cadavers showed reduced allogenicity and complete epithelization but require frequent bronchoscopies to remove exuberant granulation tissue and ultimately become malacic [34,35]. Treatments based on very low temperature freezing are effective to induce loss of class II HLA [36,37,38], and a long period of cryopreservation may help to maintain a better patency of tracheal allografts [39,40]. Cartilaginous alterations often occur [41,42,43], probably due to a deficient blood supply [44,45,46].

##### Aortic Grafts

Seven preclinical studies on a sheep model showed explored allogeneic aortic grafts as airway substitutes [47]. The aortic scaffold served as a matrix for de novo generation of cartilage. Bone marrow mesenchymal stem cells may play a role in modulating the process of regeneration [48]. This assumption led to the first human application in patients with extensive tracheal diseases [49,50,51].

De novo generation of cartilage in cryopreserved aortic allografts has been reported [12] in a small series of tracheal lesions using a cryopreserved non-AB0 matched aortic allograft supported by a stent to prevent collapse. Allografts were covered with a muscle flap to promote neovascularization and prevent fistulisation. Stents were removed 5 to 39 months after transplantation. The 90-day mortality rate was low. The regeneration of a mixed respiratory epithelium was observed on superficial allograft biopsies. De novo generation of cartilage cell of recipient within the aortic allografts was observed [52,53,54,55].

#### 2.3.2. Regenerative Medicine and Tissue Engineering

Regenerative medicine is an emerging interdisciplinary field of research and clinical applications focused on the repair, replacement, and regeneration of cells, tissues, or organs to restore damaged function from any cause, including congenital defects, diseases, and trauma [56]. This field includes therapeutic areas that were initially thought to be separate, such as cell therapy and tissue engineering (in vitro creation of tissues/organs for a subsequent transplantation as fully functional organs or tissue grafts) [57,58]. In particular, in cell therapy, the use of a biocompatible/bioabsorbable scaffold is not required, whereas in tissue engineering, a tridimensional biocompatible/bioabsorbable structure is necessary to support the regeneration of the damaged tissue [59]. These two sectors include substitution (transplantation), repair (exogenous cell therapy), or tissue regeneration (mobilization of endogenous cell pools, such as stem cells) [60,61].

In 2008, a tissue engineered tracheal graft (TETG) was implanted in a patient with severe bronchial stenosis following treatment for tuberculosis [62]. Since then, nine reports of TETG implantation in humans have been published, with a total of 15 reported patients who have undergone tracheal replacement [54,62,63,64,65,66,67,68]. The brilliant results reported by Macchiarini et al. have been made opaque from the revelation that most patients died after the implantations [52,63,68,69,70].

Tissue engineering, as a branch of regenerative medicine, has emerged as a more recent approach for restoring damaged tissues. Research in the field of tissue engineering has the objective of creating new materials that do not give rise to unwanted reactions on the part of the host organism and that are able to promote the regeneration of the damaged tissue or organ. Different strategies can be adopted [54]: insemination of cells on an implanted scaffold, implantation of tissue grown in vitro on a scaffold, or scaffold implantation without cells to support tissue regeneration in situ. Cell adhesion, proliferation, and differentiation are strongly influenced by the microenvironment as well as by the size, geometry, pore density, and the properties of the scaffold surface [61,71]. For all these approaches, the scaffold must provide a three-dimensional structure that will support the growth of a new tissue with biological properties comparable to that of the tissue that needs to be replaced [72].

The success in creating functional engineered tissues lies in the integration of cells, biomaterials, and signaling systems, also known as the tissue engineering triad [73].

The homing of stem cells into the injured area may be driven through two mechanisms: incorporation into an engineered tissue or attraction to the wound site. In the field of tracheal replacement, in an attempt to reproduce the static and dynamic features of the organ, many different scaffolds have been explored, but they can generally be classified as decellularized tracheal constructs, biosynthetic, or scaffold-free constructs.

##### Decellularized Tracheal Scaffold

The development of decellularized tracheal scaffold came in response to initial outcomes obtained using cadaveric human allografts. Decellularization provides a not immunogenic extracellular matrix (ECM) [74,75,76].

Because the mechanical properties of the tracheal ECM come from composition of collagen, glycosaminoglycans, and elastin, decellularized tracheas maintain biomechanical properties of native trachea. ECM plays an important role in tissue formation through chemoattraction, cell support, and signaling [75]. This phenomenon has been largely demonstrated by several observations of chondrocyte repopulation, re-epithelialization, muscle bundle formation, and the growth of serous glands and nerve fibers [77,78]. Decellularization protocols range from 3 days to 12 weeks to disrupt cell membranes and denature proteins [62,79,80,81,82]. Physicochemical characteristics of the decellularized scaffold promote perfusion, vascularization, and differentiation of the seeded cells. Briefly, decellularization begins with the rupture of the cell membrane by means of physical treatments or ionic solutions, followed by the separation of cellular components from the extracellular matrix through enzymatic treatments, the solubilization of the cytoplasmic components using detergents, and ends with the removal of cellular debris; these stages can be coupled with the mechanical stirring so as to increase their effectiveness. Once decellularization is complete, chemicals are removed. An effective vascular supply is fundamental for a successful tracheal transplantation: Delaere et al. [2] focused their attention on this aspect with poor results; Tan et al. [83] replaced a tract of about 5 cm of the left bronchus, removed for tumor, with a stent of Nitinol coated with porcine dermis [84]. In the study, the stent was continuously irrigated with a solution of Ringer’s lactate with added neoangiogenic factors and antibiotics. The patient survived and was discharged on month after implantation.

##### Synthetic Polymers Scaffolds and Three-Dimensional Printers

Several research groups have focused their efforts on synthetic polymers scaffolds for airway replacement. Three-dimensional printing permits to fabricate scaffold varying composition, size, and porosity of construct [85]. Copolymers of polylactic and polyglycolic acid [PLGA] as well as copolymers composed of polyphatic acid and polycaprolactone [PCL] have been tested in vitro as tubular scaffolds, to obtain a tracheal substitute. In this case, the construct was initially cellularized with mesenchymal stem cells that guaranteed complete resorption of the prosthetic material and deposited ECM [86,87,88]. Unfortunately, the process did not lead to a complete ECM formation, which is essential for the mechanical and metabolic support of the newborn tracheal tissue.

Electrospinning of soluble polymers is a promising technique for building three-dimensional porous structures [71]. Hinderer et al. [14] made a composite PCL–gelatin–decorine scaffold with a three-dimensional structure and pores of an average size of 14.4 ± 6.4 μm; the tubular scaffold was then inserted in a dynamic circuit. The results demonstrated a uniform composition of the scaffold, but a poor mechanical resistance and the presence of cells only at the outer surface of the construct. Gustafsson et al. [89] investigated the effects of culturing rat mesenchymal stromal cells on a polymeric scaffold coated with adhesion proteins. In this study, a cellular colonization was registered throughout the whole surface of the structure, regardless of the presence of adhesion molecules. Following this idea, Shi et al. [90] realized a copolymer of N-carboxyethylchitosan/nanohydroxyapatite, potentially suitable for airway replacement. However, in this first phase, the results reported from in vitro tests seem to be preliminary for in vivo application.

##### In Vivo Tracheal Scaffold Implants

Tracheal scaffolds have been also tested in vivo to evaluate tissue regeneration and healing after segmental tracheal resection. In 1994, Vacanti et al. [91] produced a tubular scaffold from sheets of fibrous polyglycolic acid cellularized with chondrocytes. The scaffolds were implanted in four rats, as substitutes for 4–6 tracheal rings. Unfortunately, the animals died soon after surgery. Kanzaki et al. [92] proposed a model of prevascularized Dacron support covered by a layer of rabbit tracheal epithelial cells. These constructs had a better outcome in a rabbit model showing mature, pseudostratified columnar epithelium that partially covered the scaffold four weeks after transplantation.

PCL was also employed to build a possible tracheal substitute. The prosthesis was tested in vivo in rabbits. The main advantages of the polymer include a slow degradation time and good mechanical characteristics [93]; unfortunately, the formation of a high amount of granulation tissue caused a stenosis in the center of the construct and, subsequently, the death of the animals.

A copolymer of lactic acid and PCL was used to produce a sponge-like tubular structure reinforced, with a PGA structure coated with gelatin, to prevent air leak, was proposed by Tsukada et al. [15]. The graft of about 6 cm in length was implanted on a sheep, performing a terminal end-to-end anastomosis and placing a silicone stent inside to avoid collapse. Nine months after implantation, the scaffold did not undergo, and the animal died consequently.

A different approach for the development of an artificial trachea was followed by Naito et al. [94]. In their experiments, fibroblast and collagen hydrogels, mechanically supported by osteogenically induced mesenchymal stem cells (MSC) in ring-shaped 3D-hydrogel cultures, were applied in a model of rat trachea injury. Tube-shaped tissue was constructed from mixtures of rat fibroblasts and collagen in custom-made casting molds. Ring-shaped tissue was constructed from mixtures of rat MSCs and collagen and fused to the tissue-engineered tubes to function as reinforcement. Unfortunately, in this case, six of the nine animals died during implantation, while three of them survived for 24 h and died the day after. Most of the tissue-engineering tracheal approaches have been focused on the development of scaffolds for circumferential defects repair. Less common scaffolds are Y-shaped substitutes for bifurcation repair. A Y-shaped scaffold made of Marlex mesh reinforced with polypropylene and coated with collagen was implanted in 20 dogs: 14 out of them died after experimentation due to obstruction, loss of air, or necrosis [95]. Huang et al. [16], on the other hand, used a PCL-based scaffold on a 47-year-old woman affected by tracheomalacia after tubercular disease. During surgery, to ensure better sealing and integration of the scaffold, it was coated with an artificial pleura patch. During the post-operative three months, the patient underwent serial bronchoscopies that allowed them to demonstrate a progressive improvement of the tracheal respiratory space (passing from 0.3 cm in maximum diameter to 1 cm). Jungebluth et al. [68] in 2011 published the results of a tracheo-bronchial transplant carried out urgently on a 37-year-old man suffering from recurrence of mucoepidermoid carcinoma involving the distal trachea. The patient had previously undergone tracheal resection and radiotherapy, but due to tumor recurrence, as a life-saving intervention, a polymer in POSS-PCU [polyhedral oligomericsilsesqui-oxane (POSS) covalently linked to poly (-carbonate-urea) urethane (PCU)], cellularized with stem cells by dynamic culture in a bioreactor, was implanted to substitute the affected tract (6 cm of distal trachea and 2 cm of the proximal main bronchi). The results obtained were good and showed a partial epithelial colonization of the polymer, as witnessed by biopsies carried out during bronchoscopic controls. After five months, the patient showed no neoplastic recurrence and a clear improvement in pulmonary function indices [96]. There are no data concerning long-term results, and there is an ongoing ethical debate on the procedure carried out [97].

##### Scaffold-Free Constructs

Scaffold-free constructs do not require cell seeding or an external three-dimensional structure by means of self-organization and self-assembly techniques.

Self-organization techniques involve the formation of new tissue with the application of external forces [17]. These techniques include bioprinting and cell-sheet engineering.

As an example, fabricated sheets of cartilage obtained from the auricular cartilage of New Zealand white rabbits have been used in combination to a muscle/silicone construct to create a neotrachea that was ectopically cultured and orthotopically transplanted 12 to 14 weeks after initial implantation [98]. Although the authors demonstrated mechanical stability without degradation, all rabbits expired due to obstruction/stenosis between 1 and 39 days after surgery.

Self-assembly techniques result in spontaneous tissue formation in the absence of external forces: cells seeded on a non-adherent surface develop neotissue by adhering to each other. Self-assembly in TETG has been reported using human MSC-derived cartilaginous rings and cylinders generated through a custom ring-to-tube assembly system [19]. Scaffold-free TETG demonstrated similar biomechanical properties when compared to native rat trachea.

##### Decellularized Constructs

Chondrocyte and epithelial cell proliferation are both needed for graft survival and patency [98]. Cell expansion can be obtained from cells seeding onto a scaffold and expanded in a bioreactor expansion prior transplant), in vivo (cells are seeded at the time of implantation with the host serving as a bioreactor), or in situ (scaffold is implanted without donor stem cells, and endogenous host cells are recruited for neotissue formation) [80].

Different MSCs have been used, ranging from bone marrow, amniotic fluid, and adipose tissue [54].

Bone marrow-derived and adipose-derived MSC have been most used, given their availability and accessibility [83].

The international patent application n. PCT/CN2014/078737 describes a method for preparing decellularized animal tissues. U.S. Patent Application No. US20150079143 describes three-dimensional scaffolds obtained by lyophilization and/or electrospinning for cell growth. International patent application no. WO2015048322 describes a method for covering bone surface with bio-mimetic polysaccharides and facilitate transplantation. Literature reports some examples of scaffolds made starting from decellularized matrix, used as such as [99], or dispersing the decellularized ECM inside a synthetic polymer. Efraim et al. [100], for example, described the development of an injectable scaffold for cardiac tissue regeneration, based on solubilization of porcine cardiac ECM combined with natural biomaterials (genipin and chitosan). It was possible to exploit the natural bioactivity and the unique structure of the cardiac tissue with the processability and modelling of the polymer.

None of these patents or works describes a complete method for covering a decellularized tissue. A partial example is reported in a study by Johnson et al. [101]: in this case, a decellularized swine trachea was reinforced with a PCL scaffold, using a 3D printer. The advantages of this methodology include the possibility to exploit and improve the mechanical properties of the decellularized tissue in order to avoid collapses; unfortunately, it is difficult to scale, and it is rather elaborate. One of the most versatile methods for the production of porous scaffolds is the phase separation induced by diffusion (DIPS) or thermally (TIPS). Scaffolds in poly-L-lactil acid (PLLA) are produced using this technique for vascular and bronchial reconstructions. The PLLA is a biodegradable, biocompatible, and bioabsorbable polymer whose degradation product is the lactic acid, already authorized by the Food and Drug Administration (FDA) for in vivo application. DIPS provides three stages: an initial coating by immersion (dip coating), one or more immersions in the coagulation bath, and finally, coating by drying [102]; in the TIPS technique, the polymer solution is cooled to the demixing temperature, cooled again at low temperature to solidify it, and finally, washed and dried. The internal dip coating can have different shapes and sizes (glass sheets for membranes [103], cylindrical supports for tubular scaffolding [104,105]). By appropriately modulating some experimental parameters, it is possible to change the characteristics of the surface, completely closed or with controlled micro-porosity. In the first case, the construct has functions of mechanical support or protection from external environment, in order to guarantee aseptic conditions; in the second case, the pores favor cell growth and differentiation, and it is suitable for the internal surfaces of the scaffold. The obtained membranes could theoretically be affixed to a decellularized tissue, but, in practice, they need to ensure adhesion to the biological substrate and/or glue adhesion with biocompatible glues. The scaffold can be foamed with PLLA on a decellularized matrix, thus forming a natural/synthetic bio-hybrid tissue, so as to exploit both the mechanical properties of PLLA and the bioactivity of the decellularized tissue during the repopulation of the scaffold, maintaining the lumen patent for all the time necessary for regeneration. A crucial and innovative aspect in the design and use of such synthetic materials is the modulation of the degradation time and of regenerative response, characterized by fibroblastic colonization, and variation of the characteristics of the biomaterial (molecular weight, porosity). The possibility of programming the resorption according to surgical needs offers significant functional advantages, with important consequences in clinical practice. In synergy with tissue growth induced by the scaffold, the use of autologous stem cells may represent an additional stimulus to the tissue regeneration process. The applicability of this model on medium-sized animals will open the door to a future human study. Of particular interest will be the study of the mechanical properties of the material, with particular reference to the possibility of being sutured, the compliance with airway pressure variations and anastomotic seal. The consequences in terms of human health are represented by a potential large-scale use of the prosthesis.

Few studies have shown regeneration of the tracheal cartilage. In fact, cartilage tissue is a low cellular tissue, and subsequently, there are few cellular elements that can be recruited. The vascularization of implanted construct is a limiting factor for the success of the implant. Therefore, in addition to the possibility of inserting chemotactic factors in the constructs to recruit progenitor cellular elements of the host, another viable path is inserting multipotent cellular elements such as induced pluripotent stem cells (IPS), capable of transforming in contact with the environment in which they are placed in tissue tracheal cartilage and, thus, favor cellular support and repopulation. IPS can be obtained from adult mesenchymal stem cell and, therefore, an available autologous source of cells [87].

Adult chondrocytes rapidly lose their phenotype in culture. Therefore, by isolating cellular elements with greater potential such as IPS [106], it offers the possibility of increasing cartilage formation. Thus, three-dimensional constructs have been created that combine the biomechanical properties of polymers with the regenerative capabilities of stem cells [107].

In Table 2, the most important attempts of tracheal reconstruction from 1994 to now are reported.

## 3. Conclusions

Many advances have been made in the field of airway reconstruction research. The ever more in-depth knowledge of the mechanisms underlying tissue regeneration with the support of bioabsorbable scaffolds seems the way that hopefully will soon give strong clinical results and, thus, make available to the thoracic surgeon useful tools to allow wider resections with less disabling outcomes for patients.

## Figures and Tables

**Table 1 life-11-00618-t001:** Tracheal reconstruction techniques.

Techniques	Methods	Features	Results
Standard tracheal reconstruction	Tracheal dilatation with rigid bronchoscope [3,6]	High recurrence rates (90%)	
	Laser surgery with placement of an endoluminal stent [9]	30–40% recurrence rate	
	Surgical resection [7,10]	Post-operative period burdened by several complications (up to 20% of cases): recurrent stenosis, permanent tracheostomy, even death	
Allografts	Tracheal allografts [11]	Early stenosis, necrosis, undergo liquefaction, and graft rejection in absence of immune suppressive therapy, radiation therapy, chemical fixation, lyophilization, and cryopreservation	Allografts need to be revascularized, cryopreserved to inhibit allogenicity and maintain structural functionality and integrity
	Cryopreserved non-AB0 matched aortic allografts [12]	Supported by a stent to prevent airway collapse and covered circumferentially with a local muscle flap to promote neovascularization	Aortic matrices played a significant role by the release of proangiogenic, chemoattractant, proinflammatory and immunomodulatory cytokines, and growth factors
Regenerative medicine and tissue engineering	Decellularized tracheal Scaffold [13]	Removal of cell from the ECM and preserving the mechanical and bioinductive profile of the graft	Breaking of cell membrane using physical treatments or ionic solutions; separation of cellular components from the extracellular matrix through enzymatic treatments; solubilization of the cytoplasmic components using detergents; removal of cellular debris
	Biosynthetic polymers Scaffolds [14,15,16]	Polyphatic acid and polycaprolactone [PCL] coated with an artificial pleura patch POSS-PCU cellularized with stem cells by dynamic culture in a bioreactor	PCL: progressive improvement of the tracheal respiratory space POSS-PCU: partial epithelial colonization of the polymer
	Scaffold-free constructs [17,18,19]	Self-organization techniques (bioprinting and cell-sheet engineering) Self-assembly techniques (cells seeded on a non-adherent surface develop neotissue by adhering to each other)	Fabricated sheets of cartilage obtained from the auricular cartilage of New Zealand white rabbits in combination with a muscle/silicone construct Self-assembly in TETG has been reported using human MSC-derived cartilaginous rings and cylinders generated through a custom ring-to-tube assembly system

**Table 2 life-11-00618-t002:** Principal tracheal reconstruction attempts from 1994 to current.

Authors	Methods	Results
Vacanti et al. [91], 1994	Tubular scaffold from sheets of fibrous polyglycolic acid cellularized with chondrocytes.	Implanted in four rats, as substitutes for 4–6 tracheal rings. The animals died soon after surgery.
Kanzaki et al. [92], 2006	Prevascularized Dacron support covered by a layer of rabbit tracheal epithelial cells.	Four weeks after transplantation, the tracheal grafts were covered by a mature, pseudostratified columnar epithelium.
Macchiarini et al. [62], 2008	A tissue engineered tracheal graft (TETG) was implanted in a patient with severe bronchial stenosis following treatment for tuberculosis.	Most patients died after the implantation of tissue-engineered airways.
Weidenbecher et al. [18], 2009	Sheets of cartilage obtained from the auricular cartilage of New Zealand white rabbits used in combination to a muscle/silicone.	Demonstrated mechanical stability without degradation but all rabbits expired due to obstruction/stenosis between 1 and 39 days after surgery.
Naito et al. [94], 2011	Fibroblast and collagen hydrogels, mechanically supported by osteogenically induced mesenchymal stem cells (MSC) in ring-shaped 3D-hydrogel cultures.	Six of the nine animals died during implantation, while three of them survived for 24 h and died the day after.
Jungebluth et al. [68], 2011	Polymer in POSS-PCU [polyhedral oligomericsilsesqui-oxane (POSS) covalently linked to poly (-carbonate-urea) urethane (PCU)], cellularized with stem cells by dynamic culture in a bioreactor carried out urgently on a 37-year-old man.	Partial epithelial colonization of the polymer.
Hinderer et al. [14], 2012	Composite PCL–gelatin–decorine scaffold with a three-dimensional structure and pores of an average size of 14.4 ± 6.4 μm.	Uniform composition of the scaffold, but a poor mechanical resistance and the presence of cells only at the outer surface of the construct.
Gustafsson et al. [89], 2012	Rat mesenchymal stromal cells cultured on a polyethylene terephthalate [PET] and polyurethane [PU] scaffold and coated with adhesion proteins.	Similar cell densities and MSC proliferating cells; no advantages with adhesion proteins.
Shi et al. [90], 2012	Copolymer of N-carboxyethylchitosan/nanohydroxyapatite chitosan/nanohydroxyapatite composites for tissue-engineered trachea.	Satisfactory tensile strength.
Huang et al. [16], 2016	PCL-based scaffold coated with an artificial pleura patch on a 47-year-old woman affected by tracheomalacia after tubercular disease.	Progressive improvement of the tracheal respiratory space (from 0.3 to 1 cm in maximum diameter).
Johnson et al. [101], 2016	In vitro characterization of design and compressive property of 3D-biofabricated/decellularized hybrid grafts for tracheal tissue engineering.	Decellularized swine trachea was reinforced with a PCL scaffold, using a 3D printer.
Tan et al. [83], 2017	Stent of Nitinol coated with porcine dermis, continuously irrigated with a solution of Ringer’s lactate with added neoangiogenic factors and antibiotics.	Patient survived and was discharged on month after implantation.
Ikeda et al. [106], 2017	Implantation of induced pluripotent stem cell-derived tracheal epithelial cells.	Survival of tracheal epithelial tissues in rat.
Hsieh et al. [108], 2018	3D printing of tubular scaffolds with elasticity and complex structure from multiple waterborne polyurethanes for tracheal tissue engineering	Stability and cartilage growth.
Chan DS [109], 2019	3D-printed polycaprolactone implants to reconstruct circumferential tracheal defects in rabbits.	Feasibility but overgrowth of granulation tissue.
Kim et al. [107], 2020	Transplantation of a 3D-printed tracheal graft combined with iPS cell-derived MSCs and chondrocytes.	Evidence in forming neocartilage.

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
