# Peer review of "Current Strategies for Tracheal Replacement: A Review"

_life, 2021, doi:10.3390/life11070618_

Round 1
Reviewer 1 Report
The topic is interesting and the article is well written. However, I have the following concerns.
Introduction
- The formula used (with descriptors) should be included in the search for the articles included in this review.
Main Body
- The author includes endoscopic dilatation and laser surgery in bronchial reconstruction, but I don't think they are strictly included in the reconstruction. Please correct the text and Table 1. If you want to leave those treatments in this text, discuss them in a separate section.
Each subtitle
- This article showed current strategies for tracheal replacement. The flow of sentences is a little difficult to understand. Is this review written in the style of a story review?
- Please add a short summary in each subtitle.
Table 2
- This table includes articles written in 1994, and 2006. These treatises are inconsistent with the heading. Please correct the heading or remove these treatises from the table.
Minor
Line 364
“In Table 2, the most important attempts of tracheal reconstruction from 2008 uo to …”. Please correct the underlined typo.
Author Response
We thank the reviewer for his advices. A brief description of method of search was included.
Regarding endoscopic dilatation and laser surgery, even if they are not reconstruction, the term was meant as reconstruction of tracheal function not only limited to anatomical continuity.
A subtitle was added for each paragraph. Minor typo error were corrected
Table 1 was added of missing references.
Reviewer 2 Report
Dear authors,
I read with great interest your work on the options for tracheal replacement. I have a few comments.
1-Could you add in the introduction data on the number of tracheal resections performed every year in Italy or in Europe, or even worldwide?
2-In table 1 concerning tracheal substitutes, there is a bar that I think is misplaced between "decellularized scaffold" and "biosynthetic polymers"
3- I would have liked to have read about the use of IPS (Induced Pluripotent Stem Cells) on decellularized scaffolds.
4-Similarly, it would be desirable to talk more about the impact of 3D printing, especially for the printing of cells in biosynthetic scaffolds
Author Response
We thank the reviewer for his comments. Minor typo error were corrected
Table 1 was added of missing references.
Total number of tracheal resection performed every year in Italy is about 30, but there is not an official public registry.
Use of IPS was included as such as the 3d printers.
Reviewer 3 Report
I have read with great interest the article and I find the subject extremely interesting.
I have the following observations:
- table 1 - lacks most of the references
- line 78 - "recurrence" of? tumor, stenosis?
- information about how the study was performed is missing: what databases were used? how were the studies selected? what search terms were used?
- my biggest concern is the fact that the last reported information is from 2016. Hasn't anything new appeared in the past 4-5 years? Is there still ongoing research with promising results or has the subject been abandoned?
I am looking forward to reading the missing information.
Author Response
A brief description of method of search was included.
Minor typo error were corrected.
Table 1 was added of missing references. Table 2 was corrected accordingly.
Round 2
Reviewer 1 Report
Table 2 shows tracheal reconstruction from 2008 to current. However, this table includes references 86 (1994) and 87 (2006). The title and contents of the table are inconsistent.
Author Response
We thank the reviewer for this advise. The table was modified and improved.
Reviewer 3 Report
Dear authors,
I have read the new version of the article and I believe that its quality has improved and some of the missing information was included.
I still have the following concerns:
- description of the search method:
- time interval is not specified (eg. last 20 years, before March 2021, or the date the databases were accessed )
- was a specific search also performed to identify clinical studies involving tracheal replacement using the ClinicalTrials.gov and University Hospital International Network Clinical trial Registry (UMIN-CTR) database? if it was done and no studies were identified I believe it would be useful to state that
- Regarding my previous concern "my biggest concern is the fact that the last reported information is from 2016. Hasn't anything new appeared in the past 4-5 years? " I believe that it was not answered properly. I have seen that a study from 2020 was added, is this the only one available? If so - please state that in the text.
I'm looking forward to reading the article
Author Response
I hope to have replied satisfactory to your questions and also I made corrections along the text. Description of the search method was modified and enhanced accordingly. Other recent studies were added as suggested.